# A Three-Pronged Approach to Studying Sublethal Insecticide Doses: Characterising Mosquito Fitness, Mosquito Biting Behaviour, and Human/Environmental Health Risks

**DOI:** 10.3390/insects12060546

**Published:** 2021-06-11

**Authors:** Mara Moreno-Gómez, Rubén Bueno-Marí, Miguel. A. Miranda

**Affiliations:** 1Henkel Ibérica S.A, Research and Development (R&D) Insect Control Department, Carrer Llacuna 22, 1-1, 08005 Barcelona, Spain; 2Laboratorios Lokímica, Departamento de Investigación y Desarrollo (I+D), Ronda Auguste y Louis Lumière 23, Nave 10, Parque Tecnológico, Paterna, 46980 Valencia, Spain; rbueno@lokimica.es; 3Área de Parasitología, Departamento de Farmacia y Tecnologia Farmacéutica y Parasitología, Facultad de Farmacia, Universitat de València, Avda. Vicent Andrés Estellés, s/n, Burjassot, 46100 València, Spain; 4Applied Zoology and Animal Conservation Research Group, University of the Balearic Islands, Cra. Valldemossa km 7,5, 07122 Palma de Mallorca, Spain; ma.miranda@uib.es

**Keywords:** prallethrin, insecticide, spatial treatment, mosquito fitness, protection, pyrethroids, *Aedes albopictus*, *Culex pipiens*, life tables

## Abstract

**Simple Summary:**

Extensive research has been carried out to assess the effects of sublethal pyrethroid doses on mosquito fitness and behaviour. Although pyrethroids are mainly used as insecticides, they can also act as repellents, depending on the dosage and/or exposure time. Females and males of two laboratory-reared mosquito species (*Culex pipiens* and *Aedes albopictus*) were exposed to five treatments in the laboratory: three doses of the pyrethroid prallethrin, as well as an untreated and a negative control. Effects on mosquito fitness, mosquito biting behaviour, and human and environmental health were evaluated. Sublethal prallethrin doses were found to decrease mosquito population size, longevity, and biting rate while posing low risks to human and environmental health. Such changes in adult mosquito fitness and behaviour could reduce the ability of mosquitoes to transmit diseases and, consequently, help limit public health risks. Although these promising results suggest sublethal insecticide doses could offer a new approach to controlling species that transmit diseases, more work is needed to identify the proper balance among regulatory requirements, contexts of usage, and human and environmental health benefits.

**Abstract:**

Worldwide, pyrethroids are one of the most widely used insecticide classes. In addition to serving as personal protection products, they are also a key line of defence in integrated vector management programmes. Many studies have assessed the effects of sublethal pyrethroid doses on mosquito fitness and behaviour. However, much remains unknown about the biological, physiological, demographic, and behavioural effects on individual mosquitoes or mosquito populations when exposure occurs via spatial treatments. Here, females and males of two laboratory-reared mosquito species, *Culex pipiens* and *Aedes albopictus*, were exposed to five different treatments: three doses of the pyrethroid prallethrin, as well as an untreated and a negative control. The effects of each treatment on mosquito species, sex, adult mortality, fertility, F1 population size, and biting behaviour were also evaluated. To compare knockdown and mortality among treatments, Mantel–Cox log-rank tests were used. The results showed that sublethal doses reduced mosquito survival, influencing population size in the next generation. They also provided 100% protection to human hosts and presented relatively low risks to human and environmental health. These findings emphasise the need for additional studies that assess the benefits of using sublethal doses as part of mosquito management strategies.

## 1. Introduction

Mosquitoes represent a major threat to human health because of their role in the transmission of vector-borne diseases (VBDs). Over the past century, the incidence of mosquito-borne diseases has increased significantly around the world [1,2,3].

To deal with this threat, researchers are developing novel techniques for use in integrated vector management (IVM) programmes and are focusing on biological, cultural, physical, mechanical, and genetic control methods [4,5]. However, chemical control, such as insecticide use, remains one of the most reliable strategies [6]. Indeed, the use of insecticides in IVM programmes has increased in recent years, reducing human mortality due to VBDs in many countries and thus playing an essential role in efforts to improve public health [7]. Pyrethroids are a key class of insecticides; they are neurotoxins that interfere with nervous system function in arthropods by blocking the closure of sodium channels. As a result, nerve impulses are prolonged, leading to muscle paralysis and, ultimately, death [8]. Worldwide, pyrethroids are the most frequently used insecticide class because they are relatively less toxic to mammals, have a rapid knockdown (KD) effect on the target arthropods, and break down rapidly in the environment due to their high degree of photodegradation [9]. They are widely deployed against agricultural pests, household pests, store-product pests, ectoparasites found on pets and livestock, and vectors of diseases [10].

Biocidal products (BPs) are strictly regulated by governmental authorities. Regulations are based on the physicochemical properties, efficacy, and environmental and human health risks posed by the active substances (ASs) contained in BPs.

Over recent decades, the European Biocidal Product Regulation (BPR) has drastically reduced the number of ASs used in insecticides, primarily as a result of toxicological and environmental concerns and, secondarily, as a result of the high costs associated with justifying the use of existing ASs or registering new ones [11]. In Europe, there are 22 official biocidal product types (PTs). The category PT18 includes the compounds used in insecticides, acaricides, and other arthropod control products that function by means other than repulsion or attraction. The category PT19 includes compounds that control harmful organisms by acting as repellents or attractants, including those that are used to protect human or animal health via spatial treatments and/or application to the skin [12]. Certain compounds, such as pyrethroids, have a dose-dependent effect: depending on the conditions of use, the substance may kill insects (PT18) [13,14] or repel them (PT19). Personal protection products can be found in both categories [13,14,15,16,17,18]. In Europe, an AS must be registered in both categories to be authorised for both uses. At present, only two ASs have such a dual status: geraniol (CAS number 106-24-1) and *Chrysanthemum cinerariaefolium* extract (CAS number 89997-63-7) [11].

EU efficacy requirements for insecticides used in space treatments stipulate that a formulation/AS dose must kill 90% of exposed insects within 24 h [19], a threshold known as the LD90. Insecticide doses below the LD90 are considered to be ineffective and, therefore, are not authorised. However, there are other issues to consider. First, high levels of mortality require the use of high doses, which conflicts with the constraints imposed by human health risk assessments (HHRAs), whose results are also required for product authorisation.

In turn, a dose is formally defined as sublethal when it induces mortality in less than 50% of exposed insects [20]. While many studies have characterised the effects of lethal pyrethroid doses on different arthropod taxa [21], much remains unknown about how sublethal pyrethroid doses used in space treatments affect mosquito fitness and behaviour or how such doses could be used in IVM programmes [18,22]. However, some studies have revealed that sublethal doses of insecticides could reduce mosquito survival, population sizes [22,23,24], and biting rates [25,26].

In this study, the effects of prallethrin 94.7% technical grade (CAS number 23031-36-9; PT18), a synthetic Type I pyrethroid, were assessed using two species of laboratory-reared mosquitoes: *Aedes albopictus* and *Culex pipiens*. Both are commonly used in insecticide efficacy tests across the globe. Prallethrin resulted in rapid knockdown (KD) when deployed against household insect pests via indoor space treatments [27]. The work presented here examined the impacts on three variables in particular: (1) mosquito fitness, (2) protection from mosquito bites in humans, and (3) toxicological risks to humans and the environment. In our analyses, we kept in mind the various constraints associated with EU authorisation standards.

## 2. Materials and Methods

The study was conducted in the Henkel Ibérica Research and Development (R&D) Insect Control Department from February 2020 to March 2021. Three experiments were performed using 5 treatments: 3 sublethal doses of prallethrin (0.40 ± 0.01 mg/h, 0.80 ± 0.01 mg/h, and 1.60 ± 0.01 mg/h), an untreated control, and a negative control.

The lowest dose, 0.4 mg/h, was used as a starting point for defining the 2 other doses. Preliminary research determined that this dose resulted in mortality rates of less than 50% 24 h after exposure (Moreno et al., unpublished data) under experimental conditions similar to those in this study (prallethrin applied via a spatial treatment in the laboratory using 12- to 14-day-old female *Ae. albopictus* and *Cx. pipiens*). Consequently, in this study, the starting dose was doubled (0.8 mg/h) and tripled (1.6 mg/h) to assess the effects of using higher levels of the AS.

To achieve accurate dosing, an electric diffuser composed of polypropylene was used (voltage = 220 V; frequency = 50 Hz; maximum power input = 10 W). It is manufactured by Henkel (model EB03) and is commercially available within the EU. The diffuser consisted of a refillable bottle containing the insecticide and a wick connected to a heater that induced evaporation. The release rate of the diffuser could be modulated by adjusting the heater temperature via the diffuser’s 2 settings. There was a normal setting, which released a minimum quantity of insecticide (mg of formula/h), and a maximum setting, which released twice that minimum quantity. Thus, to obtain a dose of 0.4 mg/h, the normal setting was used with 1.1% prallethrin in the bottle. To obtain a dose of 0.8 mg/h, the maximum setting was used with 1.1% prallethrin in the bottle. To obtain a dose of 1.6 mg/h, the maximum setting was used with 2.2% prallethrin in the bottle. Solvent types were the same in all 3 cases. The negative control used a formulation that exclusively contained the solvents. In the untreated control, mosquitoes were not exposed to prallethrin or the solvent formulation.

When the electric diffusers were not being used in the efficacy tests, they were kept running (24 h/day) in an evaporation room (temperature: 25 ± 2 °C) in the department’s chemical laboratory.

The quantities (in mg) of the formulations and the prallethrin that evaporated per hour were calculated based on the change in mass over a series of 24-h periods. Evaporation was monitored for a total of 170 h.

The experiments were carried out in a 30-m^3^ chamber, as described in Moreno et al. [28,29].

Two mosquito species—*Ae. albopictus* and *Cx. pipiens*—were used. Representatives of *Ae. albopictus* came from a colony at the Entostudio Test Institute (Italy), which Henkel has maintained for the past 8 years. Representatives of *Cx. pipiens* came from an autogenous strain that Henkel has raised at its own facilities for past 14 years; it was originally collected in the field in Barcelona (Spain). Both colonies are known to be susceptible to pyrethroids.

Mosquito-rearing conditions were as follows: a temperature of 25 ± 2 °C, a relative humidity of 60 ± 5%, and a photoperiod of 12:12 (L:D). All the experiments were conducted using 12- to 14-day-old mosquitoes. Although it is standard to estimate mortality in bioassays using mosquitoes of 5–10 days in age, older mosquitoes are more appropriate when changes in biting behaviour need to be evaluated. Thus, mosquito age was standardised for the whole study. Prior to testing, the mosquitoes were separated by species but not by sex. They were allowed to copulate but not to lay eggs. To ensure good activity levels during the experiments, the mosquitoes were given water and a 10% sucrose solution ad libitum before and during the research trials.

### 2.1. Effects of Sublethal Prallethrin Doses on Mosquito Fitness

The first experiment examined the effects of sublethal prallethrin doses on mosquito fitness and population dynamics. Female and male mosquitoes of both species were subjected to the 5 treatments. In total, 2500 mosquitoes were used: 1250 mosquitoes of each species, of which 625 were females and 625 were males. Each population of 1250 mosquitoes was divided into 10 subgroups of 125 mosquitoes. Five of the subgroups were composed of females and 5 of the subgroups were composed of males. Each subgroup was randomly assigned to 1 of the 5 treatments.

Every day, the chambers were properly cleaned and, before any experiment was begun, the chamber was checked for insecticide contamination. At least 10 mosquitoes were released into the chamber and left there for 30 min. A piece of cotton wool soaked in a 10% sugar solution was provided. Any mortality or KD during this period was noted, and the chamber was considered to be contaminated or in an unsatisfactory state if KD was higher than 10% [30]. A mosquito was considered to be KD if it was lying on its back and was unable to upright itself [31]. If no contamination was detected, the first set of mosquitoes was removed and the experimental set of 125 mosquitoes was released to initiate testing. These latter mosquitoes were given 30 min to acclimate to the chamber and were also provided with a piece of cotton wool soaked in a 10% sugar solution.

After the mosquito acclimatization period, the electric diffuser was run inside the chamber to begin the treatment. The number of mosquitoes that had been KD was counted every 10 min for up to 90 min. At the end of the trial, the mosquitoes were collected using an entomological aspirator and were taken to an insecticide-free room. There, short-term mortality (STM) was assessed at 24 h and 48 h, then long-term mortality (LTM) was assessed once a week until 100% mortality had been reached or 4 weeks had passed, whichever came first. During this period, the mosquitoes were given water and a 10% sucrose solution ad libitum. Additionally, information on locomotor impairment (i.e., loss of legs) was collected. To this end, mosquitoes were observed and classified for 48 h following a given trial. They were placed in the “living” category if they appeared to be morphologically and/or behaviourally unaffected by the treatment (i.e., they were not found lying on their backs and they had all their limbs). They were placed in the “affected” category if they had lost at least 1 leg. They were placed in the “dead” category if they were lying on their backs and failed to react to any external stimuli [32].

In addition to KD, STM, LTM, and locomotor impairment, fertility, egg laying, the ratio of females to males that emerged, and F1 population size were measured. The exact procedures differed slightly between *Cx. pipiens* and *Ae. albopictus*, as described below.

*Cx. pipiens* females: Since they came from an autogenous strain, *Cx. pipiens* females did not need to consume blood to lay eggs. Forty-eight hours after the trial, they were given a tray containing water to allow egg laying. During this period, the number of females that drowned was noted for each treatment group.*Ae. albopictus* females: Forty-eight hours after the trial, *Ae. albopictus* females were fed calf’s blood using a membrane feeding system (Hemotek, Discovery Workshops, Lancashire, England). Females were given wet paper filters for egg laying, which meant that there was no risk of drowning.

The larval rearing procedure was the same for both species. The eggs were placed in 6-L plastic trays, which were filled with 5 L of water and then labelled by treatment. The larvae developed in the trays under temperature-controlled conditions (25 °C) and were fed rat food (Nanta S.A). Larval density per tray (i.e., 100–120 larvae per litre) was carefully maintained to limit the risk of cannibalism. The water used for larva rearing was not treated with any chemical substances (i.e., anti-algal compounds). The trays were checked every day and additional food was added as needed. Upon reaching the pupal stage, individuals were transferred to the adult emergence containers.

The number of eggs laid over the course of the 4-week post-treatment period was assessed for *Ae. albopictus*, but not for *Cx. pipiens*. In the latter species, eggs are laid in groups (i.e., in egg rafts), making them difficult to count unless separated. For both species, the number of larvae that reached the third/fourth instar and the percentage of females and males that emerged were determined. The ratio of third/fourth instar larvae to females available for egg laying was also calculated.

### 2.2. Effect of Sublethal Prallethrin Doses on Mosquito Biting Behaviour

The second experiment examined the effect of sublethal doses on mosquito biting behaviour and, consequently, on host vulnerability. More specifically, it used human volunteers to determine the length of prallethrin exposure that would result in 100% protection.

Six study participants (2 men, 4 women) took part in each trial. They had undergone training to learn how to accurately count mosquito landings. Prior to testing, the skin to be exposed was washed with unscented soap, rinsed with water, rinsed with 70% ethanol or isopropyl alcohol, and then dried with an uncontaminated towel. To ensure that EU guidelines were respected, participants were asked to avoid the use of nicotine, alcohol, fragrances (e.g., perfumes, body lotions, soap), and repellents for 12 h prior to and during testing [19].

Between exposure periods, study participants remained in air-conditioned rooms and kept their activity levels low.

The trials were conducted using only non-blood-fed female *Ae. albopictus*, since the autogenous *Cx. pipiens* strain shows limited interest in feeding on humans.

To ensure good activity levels during the experiment, the mosquitoes were given water and a 10% sucrose solution ad libitum until the trial started.

As in Experiment 1, a preliminary procedure was used to check for insecticide contamination in the chamber. Once the chamber was confirmed to be clean, a pre-treatment trial took place. A total of 20 female mosquitoes were introduced into the chamber [28] and were given 30 min to acclimate. After this period, a study participant entered the chamber with the lower part of their legs exposed; the rest of their body was protected by a light beekeeper’s suit. They also wore gloves and white hospital booties [28] (Figure 1). The person remained in the chamber for 3 min [28]. During this time, the number of mosquitoes landing on their exposed skin was recorded. This figure served as a baseline for estimating percent protection following the treatment.

Percent protection expressed the relative reduction in landings/instances of probing attributable to the treatment for each participant [28]. It was calculated as follows:% protection = (C − T) × 100/C,(1)
where C = number of landings/instances of probing during the pre-treatment trial and T = number of landings/instances of probing during the treatment trial.

Immediately after the pre-treatment trial, the treatment trial began. First, the electric diffuser was switched on inside the empty chamber. After the diffuser had been running for 5 min, the person who took part in the pre-treatment trial again entered the chamber. They remained inside for 3 min, and the number of mosquitoes landing on their exposed skin was recorded. They then left the chamber. This procedure was repeated 10 min and 15 min after trial initiation.

Each participant was exposed once to each of the 3 prallethrin treatments and the 2 controls.

### 2.3. Assessments of Human and Environmental Health Risks

Toxicological risks were assessed in 2 ways: by estimating human health risks using HHRA models and by estimating environmental health risks.

HHRA models were performed for 2 populations: adults and children 2–3 years old. This work was carried out using ConsExpo Web (v. 1.0.7; [33]), a tool designed by the Dutch National Institute for Public Health and the Environment (RIVM). In ConsExpo Web, certain parameters can be set to a chosen value, while others are fixed.

Because an electric diffuser was used in the experiments, only inhalation exposure was considered. However, it is assumed that some of the AS would end up on the floor, where children 2–3 years old might be crawling, so dermal exposure in children was also considered. It was assumed that there was no oral exposure. Thus, the following ConsExpo models were used: “Inhalation exposure: exposure to spray—spray” and “Dermal exposure: direct contact with product—rubbing off”.

Within the inhalation exposure model, the inhalation rate was chosen based on Recommendation 14 of the Biocidal Product Committee (BPC) Ad Hoc Working Group on Human Exposure, which describes the default values to use when assessing human exposure to BPs [34]. In this context, here are the key values that were chosen: first, the exposure duration was 24 h per day (a worst-case scenario). Second, it was assumed that night-time respiration in the bedroom was taking place during all those hours (also a worst-case scenario). The volume of that bedroom, 16 m^3^, was one of the values fixed by ConsExpo and was considered to represent yet another worst-case scenario. To determine the exposure duration that would be considered safe for both adults and children, the 3 experimental doses were examined: 0.4, 0.8, and 1.6 mg/h (Table 1).

Within the dermal exposure model, the dislodgeable amount is the quantity of product applied on a surface area that may potentially be wiped off (per unit of surface area) and that thus may be taken up via contact between surfaces and the human skin. A worst-case scenario was assumed: 10% of the applied AS would end up on the floor, and 10% of that amount would be dislodgeable (Table 2).

To assess risks to environmental health, the following assumptions were made: continuous release (24 h/day) of a vapourised liquid containing prallethrin as its AS and the presence of 2 electric diffusers per household, as per the recommendations in the Technical Agreements for Biocides [35].

The European Chemical Agency (ECHA) Emission Scenario Document (ESD) PT18 spreadsheet (regarding indoor diffusers) was filled out in accordance with the instructions contained in the Organisation for Economic Co-operation and Development (OECD) ESD No. 18 [36]. The results were used to estimate potential product presence in wastewater following treatment and cleaning. Exposure values were calculated using the European Union System for the Evaluation of Substances (EUSES) (software v. 2.2.0).

Any additional risks resulting from metabolites were included in the risk assessment.

For each environmental compartment facing exposure, risk was characterised using the ratio of predicted environmental concentrations (PECs) to predicted no-effect concentrations (PNECs). Of greatest concern was the PEC/PNEC ratio for soils.

### 2.4. Statistical Analysis

To compare the KD and mortality curves based on species, sex, and treatment, Mantel–Cox log-rank tests including pairwise comparisons were carried out in SPSS (v. 15.0.1) for Windows (SPSS Inc., Chicago, IL, USA).

Fisher’s exact tests applying the Bonferroni correction method were used to examine treatment effects on mosquito fitness and F1 population size in *Cx. pipiens* and *Ae. albopictus.*

Generalised linear mixed models (GLMMs) were performed to determine how treatment and exposure time affected KD (Poisson error distribution and log-link function; MASS package in R) and percent protection (Gaussian error distribution and identity link function; nlme package in R). The identity of the study participant was included as a random factor. When overall significant differences were detected, pairwise comparisons were performed using *t*-tests with pooled standard deviations and the Bonferroni correction method.

The alpha level was 0.05 for all the statistical analyses.

## 3. Results

### 3.1. Effects of Sublethal Prallethrin Doses on Mosquito Fitness

In the first experiment, the following were evaluated: (1) the effects of species, sex, and treatment on KD during the 90-min treatment trial; (2) the percentage of dead and affected mosquitoes 48 h into the post-treatment period; (3) the effects of species, sex, and treatment on long-term mortality (i.e., over the 4-week post-treatment period); and (4) the effects of species, sex, and treatment on fertility, egg laying, and F1 population size.

#### 3.1.1. Effects of Species, Sex, and Treatment on KD during the 90-Min Treatment Trial

All three sublethal doses of prallethrin (0.4, 0.8, and 1.6 mg/h) caused more than 95% of mosquitoes to be knocked out, except in the case of *Cx. pipiens* females (87.2%; Figure 2). The higher the dose, the faster the KD. KD differed between the two control groups and the three prallethrin groups based on species and sex (Figure 2). In the untreated control, there was no KD. In the negative control, only a few male *Ae. albopictus* were knocked down (12.8%; Figure 2b).

First, KD was compared within species. In *Ae. albopictus*, for both sexes, there was a significant difference in KD between the mosquitoes exposed to the 0.4 mg/h prallethrin dose and the mosquitoes exposed to the 0.8 and 1.6 mg/h prallethrin doses (Table 3). Exclusively in the case of male *Ae. albopictus*, there was no significant difference between the groups exposed to the 0.8 vs. the 1.6 mg/h prallethrin dose. In general, KD was faster at the higher doses (Figure 3a,b). In *Cx. pipiens*, there were significant differences among all three prallethrin doses for both sexes (Table 3).

Second, KD was compared between species. At the lowest dose (0.4 mg/h), differences only existed between male *Ae. albopictus* and female *Cx. pipiens* (χ^2^ = 6.562, *p* < 0.05). At the intermediate dose (0.8 mg/h), male *Ae. albopictus* experienced significantly faster KD than all the other groups (*p* < 0.0001 for all the comparisons). At the highest dose (1.6 mg/h), there were no differences among female *Ae. albopictus*, male *Ae. albopictus*, and male *Cx. pipiens* (female *Ae. albopictus* vs. male *Ae. albopictus*: χ^2^ = 0.787, *p* = 0.375; female *Ae. albopictus* vs. male *Cx. pipiens*: χ^2^ = 3.645, *p* = 0.056; male *A. albopictus* vs. male *Cx. pipiens*: χ^2^ = 1.419, *p* = 0.234). However, female *Cx. pipiens* experienced significatively slower KD than all the other groups (*p* < 0.0001 for all the comparisons). For example, at 10 min, KD was only 23% for female *Cx. pipiens* but 87–92% for all the other groups (Figure 3).

#### 3.1.2. Percentage of Dead and Affected Mosquitoes 48 h into the Post-Treatment Period

Mosquitoes displayed a variety of fates during the 48 h that followed the trials. Some died, some survived, and yet others remained alive but were clearly affected by the prallethrin. The most obvious sign that surviving mosquitoes had been affected was the partial or complete loss of legs (Figure 3). This effect was observed for all the doses tested, although it was more pronounced at the higher doses (e.g., some individuals lost one or more legs and also died).

At 24 h into the post-treatment period, dead and affected mosquitoes together accounted for more than 90% of all the mosquitoes in almost all the prallethrin groups. The only exception was female *Cx. pipiens* exposed to the 0.4 mg/h prallethrin dose (41.60% at 24 h and 75.2% at 48 h).

Similarly, at 48 h into the post-treatment period, dead and affected mosquitoes together accounted for more than 90% of all the mosquitoes (females and males combined) in almost all the prallethrin groups. The only exception was *Cx. pipiens* exposed to the 0.4 mg/h prallethrin dose (84.4%).

*Dead Adult Mosquitoes*. At 24 h into the post-treatment period (Figure 4), male mortality in both species exceeded 90% in almost all the groups exposed to prallethrin. The exception was male *Cx. pipiens* exposed to the 0.4 mg/h prallethrin dose, a group that displayed 80% mortality. In both species, female mortality was lower, especially when mosquitoes were exposed to the 0.4 mg/h prallethrin dose (49.6% and 30.4% for *Ae. albopictus* and *Cx. pipiens*, respectively). At the prallethrin dose of 0.8 mg/h, female mortality was 56% for *Ae. albopictus* and 43.2% for *Cx. pipiens*. At the prallethrin dose of 1.6 mg/h, female mortality was 71.2% for both species.

At 48 h into the post-treatment period (Figure 4), the only increases in male *Cx. pipiens* mortality were seen in the groups exposed to the 0.4 and 0.8 mg/h prallethrin doses (from 80% to 84% and from 95.2% to 96.8%, respectively). Female mortality rates had risen accordingly with higher doses for both species of mosquitoes from 67.7% to 83.2% for *Ae. albopictus* and from 49.6% to 86.4% for *Cx. pipiens* (Figure 4).

*Affected Adult Mosquitoes*. At 24 h into the post-treatment period, 5% at most (range: 0.8–4.8%) of male *Ae. albopictus* were affected; the rest of the mosquitoes were dead. In the case of female *Ae. albopictus*, there were 42.4% and 40.0% affected mosquitoes in the groups exposed to the 0.4 and 0.8 mg/h prallethrin doses, respectively. At 48 h, these percentages dropped to 26.4% and 25.6%, respectively, largely because the affected mosquitoes had died. For the group exposed to the 1.6 mg/h prallethrin dose, the percentage of affected mosquitoes went from 21.6% at 24 h to 13.6% at 48 h. The same general patterns were seen in *Cx. pipiens*.

At 48 h, the percentages of affected mosquitoes were lower because mortality had occurred. For male *Cx. pipiens*, the group exposed to the 0.4 mg/h prallethrin dose had the highest percentage of affected mosquitos (13.6% at 24 h and 9.6% at 48 h). In contrast, for female *Cx. pipiens*, the percentage of affected mosquitoes increased from 11.2% at 24 h to 25.6% at 48 h for the group exposed to the 0.4 mg/h prallethrin dose; for the groups at prallethrin doses of 0.8 and 1.6 mg/h, these percentages decreased from 48.8% to 32.8% and from 26.4% to 12%, respectively.

Mortality never climbed above 15% in the untreated and negative controls, except in the case of male *Ae. albopictus* (31.2% and 32%, respectively). None of the mosquitoes in the controls showed signs of having been affected (Figure 4).

#### 3.1.3. Effects of Species, Sex, and Treatment on Long-Term Mortality

One week into the post-treatment period, total mortality for female and male *Ae. albopictus* was 90% across all the prallethrin groups; in the controls, however, total mortality was only 28%. For female and male *Cx. pipiens*, the total mortality for mosquitoes exposed to prallethrin doses of 0.4, 0.8, and 1.6 mg/h was 82%, 89.6%, and 94.8%, respectively; for the controls, it was 20.8%.

For both species and sexes, LTM was significantly higher in all the prallethrin groups than in the control groups (Table 4). Within species and sex, LTM did not differ between the untreated and negative controls; it was highest for male *Ae. albopictus* and lowest for female *Ae. albopictus* (Figure 5).

LTM did not differ between the groups exposed to the 0.4 and 0.8 mg/h prallethrin doses, regardless of species or sex. It did, however, differ between the groups exposed to the 0.4 and 1.6 mg/h prallethrin doses. It was higher at the latter dose, except in the case of male *Ae. albopictus*—they died equally rapidly across all three doses (100% mortality at 2 weeks post-treatment; Figure 5 and Table 4). In both species, male but not female LTM was significantly higher in the groups exposed to the 1.6 mg/h prallethrin dose than in the groups exposed to the 0.8 mg/h prallethrin dose (Figure 5 and Table 4).

Sex also affected mortality in the prallethrin groups: LTM was higher for males than females, regardless of species (Figure 5 and Table 4). At 2 weeks post-treatment, male mortality was higher than female mortality by 13–20% for the groups exposed to the 0.4 and 0.8 mg/h prallethrin doses and by 7–10% for the groups exposed to the 1.6 mg/h prallethrin dose.

Species-specific differences in male mortality were present at the lowest prallethrin dose: at 1 week post-treatment, male *Ae. albopictus* exhibited 99.2% mortality, while male *Cx. pipiens* exhibited 90.4% mortality (0.4 mg/h: *p* < 0.0001). There was no such difference for the intermediate prallethrin dose (0.8 mg/h: χ^2^ = 0.011, *p* = 0.918) or the highest prallethrin dose (1.6 mg/h: χ^2^ = 3.806, *p* = 0.051). Species did not affect female mortality at any of the doses (0.4 mg/h: χ^2^ = 0.826, *p* = 0.363; 0.8 mg/h: χ^2^ = 0.256, *p* = 0.613; 1.6 mg/h: χ^2^ = 0.740, *p* = 0.390).

#### 3.1.4. Effects of Species, Sex, and Treatment on Fertility, Egg Laying, and F1 Population Size over the 4-Week Post-Treatment Period

*Culex pipiens*. In this part of the experiment, the methodology diverged slightly for the two species because the *Cx. pipiens* strain did not need to consume blood (see the Methods section).

The number of eggs laid by *Cx. pipiens* could not be accurately counted because the eggs formed rafts. Furthermore, some of the rafts were not well assembled. Instead of forming the expected boat-like shape [37], unassembled eggs could be seen on the water surface (Figure 6).

Forty-eight hours after the mosquitoes had been given access to water to lay their eggs, the number of females found dead in the tray was much greater in the prallethrin groups than in the control groups (Fisher’s exact tests with Bonferroni correction: *p* < 0.001 for all the comparisons between the control groups (untreated or negative) and each of the prallethrin groups). In the control groups, fewer than 10% of females were found dead, while 23.81%, 38.78%, and 41.18% of females were found dead in the groups exposed to the 0.4, 0.8, and 1.6 mg/h prallethrin doses, respectively (Table 5).

The numbers of larvae to reach the third/fourth instar stage were similar in the untreated control (4137) and in the negative control (3985). Compared with the untreated control, the percentages of reduction in larvae that reached this development stage were 37.27%, 50.06%, and 84.60% for the groups exposed to the 0.4, 0.8, and 1.6 mg/h prallethrin doses, respectively. It is important to note that this result appeared to stem from a smaller number of adults being available to reproduce. When examining the ratio of third/fourth instar larvae to available females, there were no differences among treatments (Table 5).

The percentage of larvae reaching adulthood varied somewhat (64–74% across both sexes), although no treatment effects were observed (Fisher’s exact tests with Bonferroni correction: *p* > 0.05 for all the comparisons between treatments). The sex ratio was nearly 1:1 in the untreated control and in the group exposed to the 0.4 mg/h prallethrin dose. The sex ratio was male-biased in the groups exposed to the 0.8 mg/h and 1.6 mg/h prallethrin doses.

There was a pronounced effect of treatment on the F1 population size. Using the untreated control as the standard of comparison, exposure to the 0.4, 0.8, and 1.6 mg/h prallethrin doses reduced the F1 population sizes by 31.25%, 53.13%, and 84.13%, respectively. Declines in population size were significatively different among the three prallethrin groups (Fisher’s exact tests with Bonferroni correction: *p* < 0.005 for all the comparisons).

*Aedes albopictus*. The same data were collected for *Ae. albopictus*, but, in addition, egg number was quantified. As the eggs were laid on wet filter paper, females were not at risk of drowning. In all the groups, including controls, the percentage of females found dead in the egg-laying trays was less than 1%, except for the group exposed to the 0.8 mg/h prallethrin dose (5.41%) (Table 6).

When examining the ratio of third/fourth instar larvae to available females, no consistent pattern was seen. While there were 15.59 larvae for each female in the group exposed to the 0.4 mg/h prallethrin dose, this figure was 3.86 and 8.24 in the groups exposed to the 0.8 and 1.6 mg/h prallethrin doses, respectively. A difference was also observed between the controls (untreated control: 13.20 larvae to 1 female; negative control: 9.44 larvae to 1 female; Table 6).

The percentage of larvae reaching adulthood (75–99%) displayed no treatment effects (*p* > 0.05), except the group exposed to the 0.8 mg/h prallethrin dose that differed from the other two prallethrin groups (*p* < 0.00001). The sex ratio was biased towards females, ranged from 0.7 to 1.0, and was unaffected by the treatments.

There was again a pronounced effect of treatment on the F1 population size. Population size declined by 32.95%, 60.6%, 91.55%, and 89.94% in the negative control group and in the groups exposed to the 0.4, 0.8, and 1.6 mg/h prallethrin doses, respectively. Dose significantly affected declines in population size in almost all cases (Fisher’s exact tests with Bonferroni correction: *p* < 0.00001 for all the comparisons except that between the groups exposed to the 0.8 versus the 1.6 mg/h dose (*p* > 0.05)) (Table 6).

### 3.2. Effects of Sublethal Prallethrin Doses on Mosquito Biting Behaviour

Percent protection after 5 min of exposure ranged from 80.07% (±28.38) at the 0.4 mg/h dose to 100% at the 1.6 mg/h dose, but this difference was not significant (*p* > 0.05); (Figure 7. The control treatments provided no protection. At this same time point, KD was null for the two controls; it was 9.33% (±5.39), 17.67% (±49.62), and 51.67% (±7.44) for the 0.4, 0.8, and 1.6 mg/h prallethrin doses, respectively. No significant differences were observed in KD between the groups exposed to the 0.4 versus the 0.8 mg/h dose (*p* > 0.05); there were significant differences in KD at 5 min for the groups exposed to the 0.4 versus the 1.6 mg/h dose and the 0.8 versus the 1.6 mg/h dose (*p* < 0.00001 in both cases). After the diffuser had been running for 15 min, 100% protection was seen in all the prallethrin groups (*p* > 0.05). KD remained null for the two controls; it was 80.17% (±10.25), 95.83% (±4.92), and 100.00% (±0.00) for the 0.4, 0.8, and 1.6 mg/h prallethrin doses, respectively (Figure 7). There was a significant difference between the groups exposed to the 0.4 versus the 1.6 mg/h dose (*p* < 0.05) but not between the groups exposed to the 0.4 versus the 0.8 mg/h dose (*p* > 0.05) or the groups exposed to the 0.8 versus the 1.6 mg/h dose (*p* > 0.05). 

When assessing percent protection, there were no differences between the untreated and negative controls at any of the time points (i.e., *p* > 0.05 at all time points). The same pattern was seen for KD (*p* > 0.05 at all time points).

When the relationship between KD and percent protection was examined, it was found that once KD reached 10%, protection never dropped below 90%. In the controls, negative percent protection values were observed because there were greater numbers of landings during the treatment trial than during the pre-treatment trial. KD was not observed in the control groups (Figure 8).

### 3.3. Assessments of Human and Environmental Health Risks

The HHRA models found that if a prallethrin dose of 1.6 mg/h were to be used, adults could be exposed for 24 h per day, but children could only safely be exposed for 12 h per day. At a prallethrin dose of 0.8 mg/h, children could be exposed for a maximum of 20 h per day. At the lowest dose, 0.4 mg/h, both adults and children could be exposed for 24 h per day.

In the environmental risk assessment, PECs and PNECs were determined for different environmental compartments. When the PEC/PNEC ratio is greater than 1, the AS poses a risk. If prallethrin were to be used 24 h per day and released using two diffusers per household, it would not be safe to use a dose of 1.6 mg/h (PEC/PNEC ratio for soils: 1.34). However, lower doses—0.8 and 0.4 mg/h—would be safe under the same usage conditions (PEC/PNEC ratio for soils: 0.75 and 0.33, respectively).

## 4. Discussion

When used at sublethal doses applied via a diffuser-mediated spatial treatment, the pyrethroid prallethrin affected the fitness of laboratory-reared *Cx. pipiens* and *Ae. albopictus* adult mosquitoes. The insecticide influenced short- and long-term mosquito mortality, physical status, and egg laying. As a result of reduced mosquito fitness, the size of the F1 population declined in the three prallethrin groups in both species. The mosquitoes’ behaviour was also altered. Biting was completely inhibited in as little as 15 min, offering 100% protection to potential human hosts. The modelling revealed that lower doses pose less risk to human and environmental health.

More than 50% of female mosquitoes were still alive 24 h after exposure to the 0.4 and 0.8 mg/h prallethrin doses; this figure was 28.8% for the 1.6 mg/h prallethrin dose. Although technically alive, these mosquitoes nonetheless suffered severe damage to their locomotor systems (e.g., they were missing up to five legs; Figure 4). Previous studies have also observed this phenomenon in response to insecticide exposure [38,39]. Leg loss could theoretically have a major impact because mosquitoes use their legs for a wide variety of functions, including locomotion, mechanical support (e.g., remaining on the water surface, laying eggs), chemical communication, sensory perception of the environment, and protection from desiccation [40,41]. However, other work found that insecticide-induced leg loss did not significantly affect the success of blood feeding or egg laying [38]—mosquitoes with fewer legs were still able to bite humans and reproduce, maintaining their life cycle. The mortality of adult mosquitoes increased in the days following prallethrin exposure, a pattern that may have been due, entirely or in part, to the insecticide’s irreversible effects on the nervous system. For example, the mosquitoes may have been unable to metabolise the AS [42], or they may have struggled to seek out and/or acquire food [43]. Furthermore, female *Cx. pipiens* were found dead in the water when eggs were counted at 48 h post-treatment. It may be that, having lost legs, they were unable to remain on the water surface when laying eggs [38,44]. The combined percentage of dead and affected mosquitoes exceeded 90% for almost all groups at 24 h into the post-treatment period. The only exception was the female *Cx. pipiens* exposed to the 0.4 mg/h prallethrin dose (24 h: 41.6% and 48 h: 75.20%). According to European efficacy guidelines, for an AS/BP to be officially classified as an insecticide useable in spatial treatments, it must kill 90% of females within 24 h of exposure [30]. None of the doses tested in this study would meet the minimum requirements allowing insecticide authorisation; repellent use would also be prohibited because the compound is not authorised for that purpose. It should be noted that the 24-h window of observation means that authorisation decisions are based solely on “immediate” mosquito mortality. Therefore, the long-term mortality observed in this study would not be taken into account for authorisation purposes, even if the mosquitoes were to be “moribund/affected” at 24 h and then finally die at 48 h [30]. OECD guidelines provide specific instructions for such situations: “*Insects in [a] supine position and those [in a] ventral position without [the] ability to move forward and exhibiting uncoordinated or sluggish movements of legs are classified as moribund. Moribund test organisms are counted as dead, if they die within the test duration*” [32].

Looking at the long-term mortality, starting at 1 week into the post-treatment period, total mortality (females and males) for both species for all the prallethrin doses was 80–95%. The lowest level of LTM, 82.4%, was seen in the *Cx. pipiens* exposed to the 0.4 mg/h prallethrin dose. The highest level of LTM, 94.8%, also occurred in *Cx. pipiens*, in the mosquitoes exposed to the 1.6 mg/h prallethrin dose. In contrast, in the controls, total LTM was lower than 30% for both species. At the end of the first experiment (i.e., 4 weeks into the post-treatment period), even doubling the dose from 0.4 to 0.8 mg/h did not significantly increase LTM, regardless of species or sex. However, LTM did climb when tripling the dose from 0.4 to 1.6 mg/h. It should be noted that the mosquitoes in all the prallethrin groups had significatively higher LTM than the mosquitoes in all the control groups (Figure 1); there was no difference in LTM between the untreated and negative controls. Additionally, the first experiment showed that females were less susceptible than males to prallethrin (Figure 5). Sex-specific differences in susceptibility to insecticides have been seen before in laboratory populations [45] and field populations [46]. In both cases, males were found to be more susceptible than females. It is hypothesised that this difference is related to the males’ smaller size and/or greater physiological susceptibility [47,48]. Nevertheless, it should be noted that, in all treatments, females survived significantly longer than did males. Consequently, biological factors appear to also influence mosquito mortality and survival.

Prallethrin exposure caused a marked decline in the size of the F1 population. The higher the dose, the larger the decline, which reached a maximum of 80–90% for both species. The above pattern likely stemmed from the higher mortality in exposed mosquitoes. The insecticide did not appear to affect female fertility in *Ae. albopictus*, given that, across treatment groups, there was consistency in the ratio of larvae to females (see Table 6). Additionally, because eggs could be accurately counted in this species, it was possible to confirm that the percentage of eggs that developed into third/fourth instar larvae was also fairly consistent (43.36% in the negative control and 53.8% for mosquitoes exposed to the 0.4 mg/h prallethrin dose), although it was rather low for the group exposed to the 0.8 mg/h prallethrin dose. For *Cx. pipiens*, it was hypothesised that insecticide exposure could affect egg viability via its impacts on raft assemblage (Figure 7) [37]. This hypothesis was based on the results of previous research. For example, Bibbs et al. [22] discovered that sublethal doses of the pyrethroid transfluthrin could cause chorion collapse in *Ae. aegypti* eggs, rendering them non-viable. In this study, the eggs of *Ae. albopictus* did not show any external signs of damage that could suggest issues with their viability. However, no clear conclusions could be drawn from the ratio of larvae to females, which ranged between 35.27 for the untreated control and 42.16 for the mosquitoes exposed to the 0.8 mg/h prallethrin dose.

Other studies have shown that exposure to pyrethroid vapours (i.e., those of metofluthrin or transfluthrin) at sublethal doses can affect female fertility and egg laying by causing declines in egg viability [22,24] and larval survivorship [24]. However, in those studies, the mosquitoes were placed in small containers (<500 cm^3^), not in a large chamber as in this study (30 m^3^). Room size and/or the distance of the mosquitoes from the source of the insecticide could influence treatment efficacy. Another factor that could have an influence on the results is whether the mosquitoes were free flying or in cages. For example, any equipment used to constrain the mosquitoes could restrict the aerial diffusion of the AS [15,23,49]. Here, mosquitos could fly freely within a large chamber. As a result, it was impossible to control mosquito distance from the diffuser, but such a design probably better replicates AS use in real life and their influence on mosquitoes. Thus, returning to this study’s results, the testing conditions used did not allow clear conclusions to be made about the effect of sublethal prallethrin doses on mosquito fertility. Further research is needed to determine whether more prolonged prallethrin exposure (i.e., longer than 90 min) could yield more definitive results.

With regards to biting behaviour, even the lowest dose of prallethrin, 0.4 mg/h, reduced the host-seeking efficiency of mosquitoes, resulting in 100% protection and 80–100% KD after 15 min. However, it was not necessary to reach 80% KD to greatly inhibit biting (Figure 8). In fact, even when just 10% of the population was knocked down, the level of protection against mosquito bites was approximately 90% (Figure 8). This result can be explained by prallethrin’s effects. At low doses/exposure times, the insecticide causes mosquitoes to become disoriented. At higher doses/exposure times, the effects on the nervous system are more pronounced. Certain mosquitoes are knocked down, while others experience a dramatic impairment of their host-seeking abilities [50,51]. Although the importance of modifying vector behaviour has been recognised for decades, the utility of this tool remains greatly underestimated from the standpoints of both BP authorisation and disease control efforts.

When assessing an AS, it is also crucial to consider any risks to human and environmental health. The toxicological results showed that only the lowest dose (0.4 mg/h) would allow 24-h insecticide use by adults and children indoors while also limiting the environmental risks. However, such a low dose would not be authorised in this context of use under current EU requirements for insecticides, which only focus on immediate mortality and do not consider additional data such as LTM and/or beneficial behavioural modifications. Further studies are needed to define how much longer exposure would need to last at low doses for the compound to meet European efficacy requirements (i.e., 90% mortality within 24 h).

Worldwide, pyrethroids are commonly used to control insects, both at the individual level and the environmental level; for example, they are frequently part of IVM programmes [52]. Extensive research has been carried out to assess the effects of sublethal pyrethroid doses on mosquito fitness [22,24,49] and behaviour [23,53,54]. Although pyrethroids are used as insecticides, they can also function as repellents when certain doses or exposure times are used. If insecticides have appropriate levels of volatility, they can be used in space treatments at sublethal doses. Examples of such insecticides include metofluthrin [24,49], transfluthrin [22,55], d-allethrin [25], or prallethrin, the compound studied here [54]. Less volatile insecticides such as permethrin or deltamethrin function better as contact repellents [26,56,57]. For the latter group to be effective, mosquitoes must come into direct contact with the AS, which is possible when insecticides are applied to netting, for example [58,59]. In the case of space treatments, mosquitoes can detect the airborne compounds and avoid entering the treated area [18,60,61]. Multiple studies have demonstrated the efficacy of these insecticides at low doses and their potential benefits for public health and mosquito control efforts [22,23,24,25,49,60]. However, in Europe, they are only authorised for use as insecticides, which greatly limits their potential utility [11].

This study found that sublethal prallethrin doses applied indoors via a spatial treatment had a significant effect on mosquito mortality and biting behaviour. This approach could thus potentially be used to reduce the vector capacity of mosquitoes and, consequently, public health risks. Although the research results presented here are promising, more studies on this complex topic are obviously needed. First, this study utilised two mosquito strains that have been bred exclusively in the laboratory for several years. As a result, it is unknown how well the above findings may reflect the reality in wild mosquito populations. Further studies addressing this issue should be performed. There are other directions that future research can take to explore the benefits and/or limitations of using sublethal doses of pyrethroids in mosquito control efforts. A logical tack to take is to further examine the usefulness of sublethal pyrethroid doses in IVM programmes by evaluating how compounds used as spatial treatments operate under field conditions. Although the concentration of the AS in the air is much lower, the environmental risks could be greater. When considering outdoor applications, an important factor to examine is the development of resistance in mosquito populations via continuous exposure to sublethal pyrethroid doses. Potential shifts in vector sensitivity or susceptibility under such conditions must be explored to assess the likelihood of this potential side effect [62,63,64].

It is essential to remember that, in the future, a major constraint will be the costs associated with justifying the use of, evaluating the efficacy of, and registering new compounds or compound uses under the BPR [65]. By utilising new evaluation parameters and/or adopting new authorisation paradigms (i.e., LTM and mosquito biting behaviour), it should be possible to exploit currently authorised compounds in new ways [66]. As a result, it may be possible to eliminate the above barrier to innovation and thus help ensure the continued availability of compounds that can effectively control mosquitoes while limiting risks to human and environmental health.

## Figures and Tables

**Figure 1 insects-12-00546-f001:**
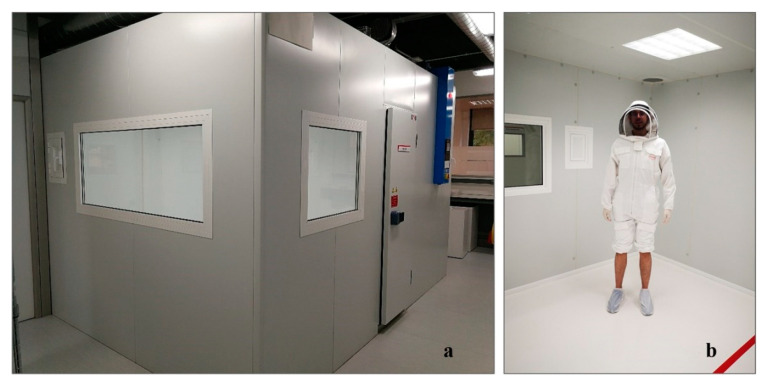
(**a**) The 30-m^3^ testing chamber at Henkel’s R&D Laboratory. (**b**) Participant wearing a protective suit while inside the chamber.

**Figure 2 insects-12-00546-f002:**
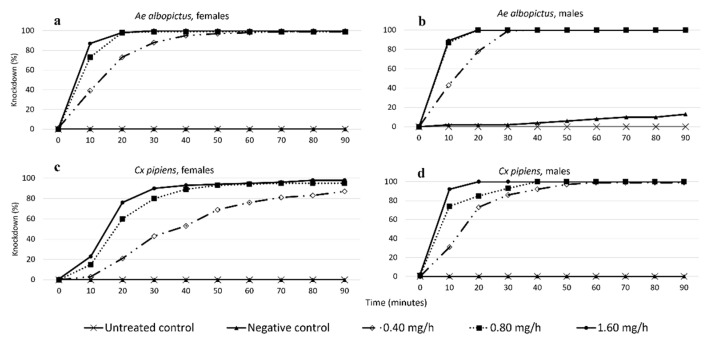
Knockdown over the 90-min treatment trial in Experiment 1 for female and male *Ae. albopictus* and *Cx. pipiens* across the five treatment groups: (**a**) Female *Ae. albopictus*, (**b**) male *Ae. albopictus*, (**c**) female *Cx. pipiens*; and (**d**) male *Cx. pipiens*.

**Figure 3 insects-12-00546-f003:**
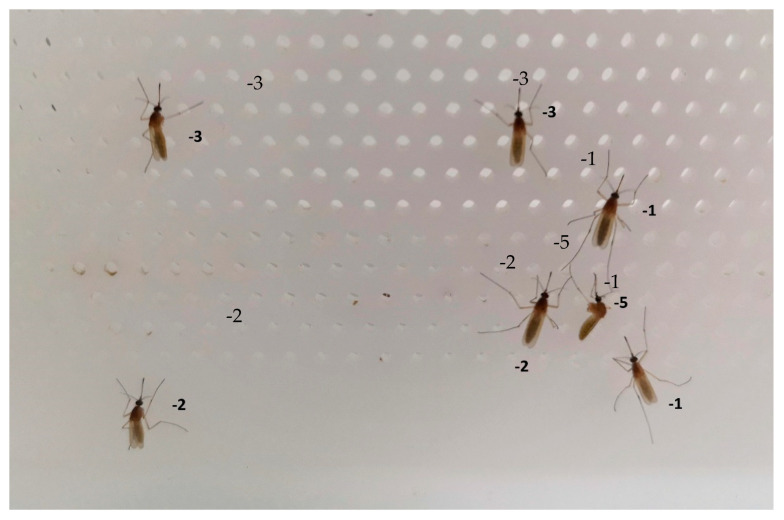
Photograph showing a sample of female *Cx. pipiens* that lost legs following prallethrin exposure. The numbers next to the mosquitoes indicate the number of legs lost.

**Figure 4 insects-12-00546-f004:**
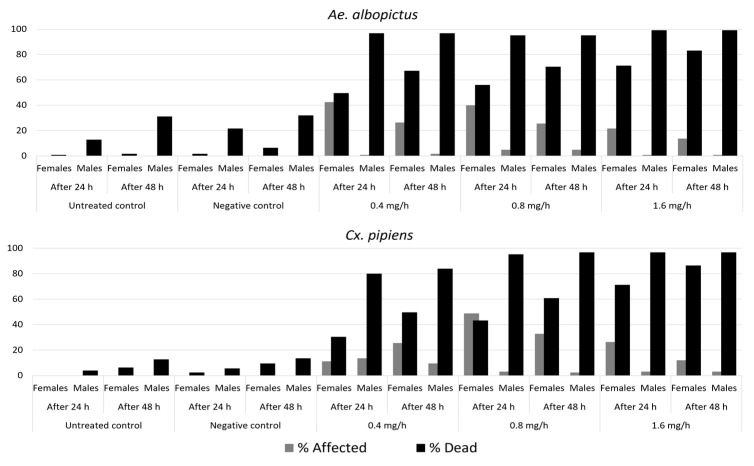
Percentages of affected and dead *Ae. albopictus and Cx. pipiens* at 24 and 48 h into the post-treatment period across the five treatment groups.

**Figure 5 insects-12-00546-f005:**
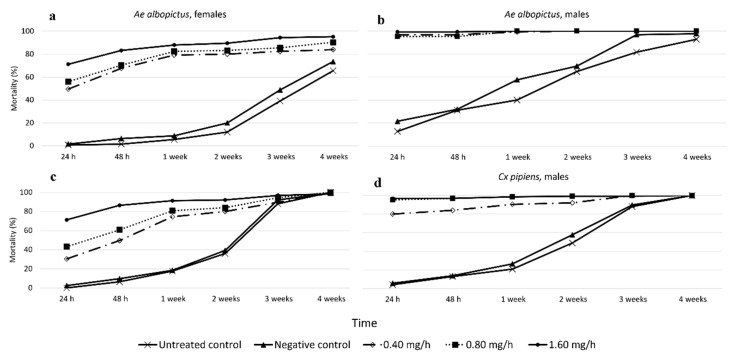
Mosquito mortality during the 4-week post-treatment period across the five treatment groups: (**a**) Female *Ae. albopictus*, (**b**) male *Ae. albopictus*, (**c**) female *Cx. pipiens*, and (**d**) male *Cx. pipiens*. Mortality at 24 h and 48 h is also shown to clarify the relationship between STM and LTM. LTM, long-term mortality; STM, short-term mortality.

**Figure 6 insects-12-00546-f006:**
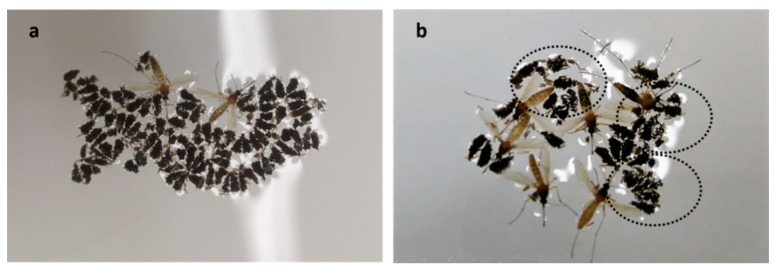
Egg rafts produced by *Cx. pipiens* in the (**a**) untreated control group and (**b**) the group exposed to the 0.8 mg/h prallethrin dose. In (**b**), the poorly assembled egg rafts have been circled to make them easier to identify.

**Figure 7 insects-12-00546-f007:**
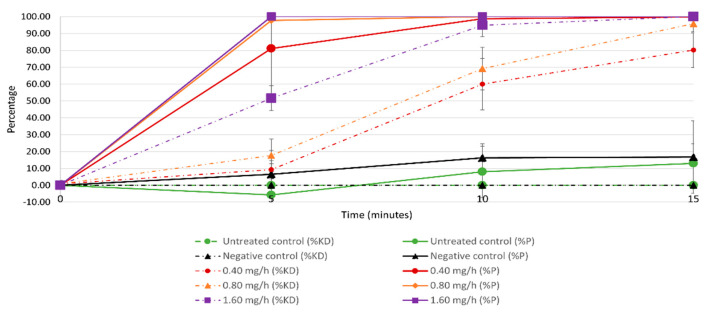
Percent protection (%*p*) and knockdown (%KD) over time for *Ae. albopictus* across the five treatment groups in Experiment 2.

**Figure 8 insects-12-00546-f008:**
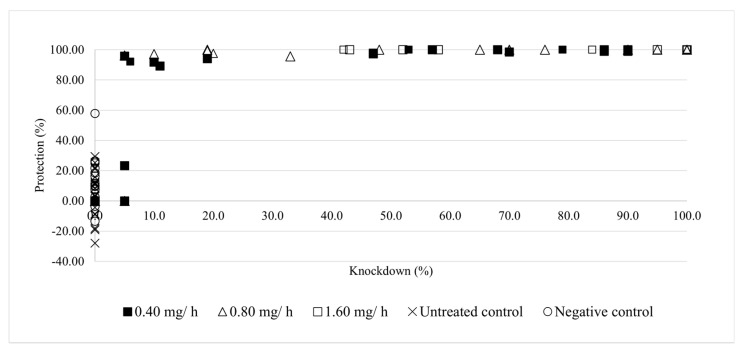
Relationship between knockdown and percent protection for *Ae. albopictus* across the five treatment groups in Experiment 2.

**Table 1 insects-12-00546-t001:** Summary of parameters for the ConsExpo model “Inhalation exposure: exposure to spray—spray”.

Parameter	Value
Spray duration	24 h (worst-case scenario)
Exposure duration	To be determined (max. number of hours that exposure remained safe for adults and children)
Weight fraction compound	100% (the prallethrin release rate is considered in the mass generation rate)
Room volume	16 m^3^ (fixed value)
Room height	2.5 m (fixed value)
Ventilation rate	1/h (fixed value)
Inhalation rate	16 m^3^/d (adult)
	10.1 m^3^/d (child of 2–3 years old)
Mass generation rate	4.03 × 10^−5^ g/s (=1.6 mg/h)
	2.27 × 10^−5^ g/s (=0.8 mg/h)
	1.02 × 10^−5^ g/s (=0.4 mg/h)
Airborne fraction	1 (fixed value)
Density, non-volatile	0.85 g/cm^3^ (density corrected to formulation)
Inhalation cut-off diameter	15 µm (fixed value)
Aerosol diameter distribution	log normal (fixed value)
Median diameter	8 µm (fixed value)
Coefficient of variation	0.3 (fixed value)
Maximum diameter	50 µm (fixed value)
Body weight	60 kg (adult), 15.6 kg (child 2–3 years old)
Absorption	100% (fixed value)

Chosen and fixed parameter values for the ConsExpo model [33].

**Table 2 insects-12-00546-t002:** Summary of parameters for the ConsExpo model “Dermal exposure: direct contact with product—rubbing off”.

Parameter	Value
Weight fraction compound	100% (the prallethrin release rate is considered in the mass generation rate)
Transfer coefficient ^1^	0.24 m^2^/h (fixed value)
Dislodgeable amount	2.93 mg/m^2^
Contact time	60 min (fixed value)
Rubbed surface	7 m^2^ (fixed value)
Absorption model	Fixed fraction
Absorption	6% (based on experimental results provided by the AS supplier)

AS, active substance. ^1^ Chosen and fixed parameter values for the ConsExpo model [33].

**Table 3 insects-12-00546-t003:** Comparison of within species knockdown for female and male *Ae. albopictus* and *Cx. pipiens* across the five treatment groups in Experiment 1.

Species	Sex	Treatment Comparisons	χ^2^	*p*-Value
*Ae. albopictus*	Females	Untreated vs. negative control ^1^	-	-
		Controls vs. prallethrin groups ^2^	-	*p* < 0.0001 in all cases
		0.4 mg/h vs. 0.8 mg/h	34.59	*p* < 0.0001
		0.4 mg/h vs. 1.6 mg/h	63.02	*p* < 0.0001
		0.8 mg/h vs. 1.6 mg/h	6.18	*p* < 0.05
	Males	Untreated vs. negative control	17.03	*p* < 0.0001
		Controls vs. prallethrin groups ^2^	-	*p* < 0.0001 in all cases
		0.4 mg/h vs. 0.8 mg/h	61.76	*p* < 0.0001
		0.4 mg/h vs. 1.6 mg/h	65.21	*p* < 0.0001
		0.8 mg/h vs. 1.6 mg/h	0.15	*p* = 0.698
*Cx. pipiens*	Females	Untreated vs. negative control ^1^	-	-
		Controls vs. prallethrin groups ^2^	-	*p* < 0.0001 in all cases
		0.4 mg/h vs. 0.8 mg/h	39.88	*p* < 0.0001
		0.4 mg/h vs. 1.6 mg/h	67.29	*p* < 0.0001
		0.8 mg/h vs. 1.6 mg/h	5.49	*p* < 0.05
	Males	Untreated vs. negative control	-	-
		Controls vs. prallethrin groups ^2^	-	*p* < 0.0001 in all cases
		0.4 mg/h vs. 0.8 mg/h	25.28	*p* < 0.0001
		0.4 mg/h vs. 1.6 mg/h	102.49	*p* < 0.0001
		0.8 mg/h vs. 1.6 mg/h	22.23	*p* < 0.0001

Pairwise comparisons of knockdown (KD) were carried out using Mantel–Cox log-rank tests in implemented in in SPSS (v. 15.0.1) for Windows (Chicago, SPSS Inc). All the statistical comparisons used an alpha level of 0.05. ^1^ No statistics were performed because no mosquitoes were knocked down in the controls. ^2^ Each control group (untreated and negative) was compared with each prallethrin group (0.4, 0.8, and 1.6 mg/h). This row summarises the results. Significant differences were observed between the control groups and the prallethrin groups in all the configurations.

**Table 4 insects-12-00546-t004:** Treatment effects on long-term mortality for female and male *Ae. albopictus* and *Cx. pipiens* across the five treatment groups.

Species	Sex	Treatment Comparisons	χ^2^	*p*-Value
*Ae. albopictus*	Females	Untreated vs. negative control	3.15	*p* = 0.07
		Controls vs. prallethrin groups ^1^	-	*p* < 0.0001 in all cases
		0.4 mg/h vs. 0.8 mg/h	0.15	*p* = 0.69
		0.4 mg/h vs. 1.6 mg/h	6.40	*p* < 0.05
		0.8 mg/h vs. 1.6 mg/h	5.72	*p* < 0.05
	Males	Untreated vs. negative control	6.32	*p* < 0.05
		Controls vs. prallethrin groups ^1^	-	*p* < 0.0001 in all cases
		0.4 mg/h vs. 0.8 mg/h	0.06	*p* = 0.80
		0.4 mg/h vs. 1.6 mg/h	2.07	*p* = 0.14
		0.8 mg/h vs. 1.6 mg/h	3.66	*p* = 0.056
*Cx. pipiens*	Females	Untreated vs. negative control	3.15	*p* = 0.07
		Controls vs. prallethrin groups ^1^	-	*p* < 0.0001 in all cases
		0.4 mg/h vs. 0.8 mg/h	0.15	*p* = 0.69
		0.4 mg/h vs. 1.6 mg/h	6.40	*p* < 0.05
		0.8 mg/h vs. 1.6 mg/h	5.72	*p* < 0.05
	Males	Untreated vs. negative control	1.48	*p* = 0.22
		Controls vs. prallethrin groups ^1^	-	*p* < 0.0001 in all cases
		0.4 mg/h vs. 0.8 mg/h	0.93	*p* = 0.33
		0.4 mg/h vs. 1.6 mg/h	4.69	*p* < 0.05
		0.8 mg/h vs. 1.6 mg/h	2.14	*p* = 0.14

^1^ Each control group (untreated and negative) was compared with each prallethrin group (0.4, 0.8, and 1.6 mg/h). This row summarises the results. Significant differences were observed between the control groups and the prallethrin groups in all the configurations. Pairwise comparisons of long-term mortality (LTM) were carried out using Mantel–Cox log-rank tests implemented in SPSS (v. 15.0.1) for Windows (SPSS Inc., Chicago, IL, USA). All the statistical comparisons used an alpha level of 0.05.

**Table 5 insects-12-00546-t005:** Treatment effects on mosquito fitness and F1 population size in *Cx. pipiens*.

Variables Measured		Untreated Control	Negative Control	0.4 mg/h	0.8 mg/h	1.6 mg/h
No. of females alive after 48 h		117	113	63	49	17
% of females found dead in the egg laying tray		8.55	9.73	23.81	38.78	41.18
No. third/fourth instar larvae		4137	3985	2595	2066	637
Ratio of larvae/females		35.36	35.27	41.19	42.16	37.47
% larvae reaching adulthood	Males	36	ND	35.8	39.8	39.7
	Females	32.1	ND	38.8	24.1	30.5
Total no. of adults in F1 population		2816	ND	1936	1320	447
% reduction in F1 population size ^1^		-	ND	31.25	53.13	84.13

ND, no data. In the negative control, algae began growing in some of the trays, creating a surface layer that choked off a large percentage of the larvae. This portion of the experiment thus had to be stopped for this group. ^1^ This metric was calculated for the prallethrin groups based on the total number of adults in the F1 population in the untreated control.

**Table 6 insects-12-00546-t006:** Treatment effects on mosquito fitness and F1 population size in *Ae. albopictus*.

Variables Measured		Untreated Control	Negative Control	0.4 mg/h	0.8 mg/h	1.6 mg/h
No. of females alive after 48 h		123	117	41	37	21
% of females found dead in the egg laying tray		0.81	0	0	5.41	0
No. eggs laid		3434	2525	1187	508	356
No. third/fourth instar larvae		1624	1104	639	143	173
Ratio of larvae/females		13.20	9.44	15.59	3.86	8.24
% larvae reaching adulthood	Males	37.32	37.77	33.80	41.26	37.57
	Females	42.86	41.30	46.32	58.04	38.15
Total no. of adults in F1 population		1.302	873	512	110	131
% reduction in F1 population size ^1^		-	32.95	60.68	91.55	89.94

^1^ This metric was calculated for the prallethrin groups based on the total number of adults in the F1 population in the untreated control.

## Data Availability

The datasets generated during and/or analysed during the study are available from the corresponding author upon reasonable request.

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
