# Peer review of "A Three-Pronged Approach to Studying Sublethal Insecticide Doses: Characterising Mosquito Fitness, Mosquito Biting Behaviour, and Human/Environmental Health Risks"

_insects, 2021, doi:10.3390/insects12060546_

Round 1

Reviewer 1 Report

The manuscript describes an innovative study of sublethal effects of pyrethroid insecticides on Aedes aegypti and puts results in context with respect to host protection from biting and toxicological aspects of host exposure. The study is well-written and clearly presented. However, more detail is required in the methods section and there is only minimal statistical analysis of the results and this section needs improvement. I have a few suggestions for further development of the manuscript:

Methods:

  1. Please give more detail about the prallethrin used (technical grade or formulated product?, manufacturer, Type I pyrethroid)
  2. Are technical details and manufacturer information available for the diffuser used?
  3. Long-term laboratory colonies of both species of mosquitoes were used. Are these colonies outcrossed to wild mosquitoes at specific intervals? If not, please comment on their fitness compared with wild mosquitoes and whether the long-term rearing is likely to have had any effect on the results of the study.
  4. Are commercial details (product name) and manufacturer available for the rat food used to feed the larvae?
  5. What was the larval density used?
  6. Line 209 - When first referring to protection, it would be best to specify that it is protection from biting that you are looking at.
  7. Line 229 - Did an external observer count the number of mosquitoes landing or was this done by the person being exposed?

Results and Discussion

  1. Is leg loss necessarily due to over stimulation of the nervous system? Is there a reference to back up this idea? Leg autotomy has been described in moths as a way of reducing the dose of insecticide received (see Aubrey Moore, Bruce E. Tabashnik, Leg Autotomy of Adult Diamondback Moth (Lepidoptera: Plutellidae) in Response to Tarsal Contact with Insecticide Residues, Journal of Economic Entomology, Volume 82, Issue 2, 1 April 1989, Pages 381–384, https://doi.org/10.1093/jee/82.2.381). There is also research on mosquitoes in this area which is of relevance to this paper which would be worthwhile discussing and citing (see Isaacs, A. T., Lynd, A., & Donnelly, M. J. (2017). Insecticide-induced leg loss does not eliminate biting and reproduction in Anopheles gambiae mosquitoes. Scientific reports, 7, 46674. https://doi.org/10.1038/srep46674)
  2. I see no statistical comparisons for data in Tables 5 and 6.
  3. Please comment on the statistical power of the exposure/protection experiment. Are statistical comparisons of the data possible?

Author Response

We would like to thank the reviewers for their extremely constructive criticism, and we have tried to incorporate their feedback to the greatest degree possible. Within the manuscript, any changes are underlined and highlighted in red.

Authors Response please see the attached.

Reviewer 2 Report

GENERAL COMMENTS

It seems like one of the arguments the authors are putting forth in this manuscript is that the EU should consider authorising active substances that, while not killing 90% or more of exposed insects within 24 hours, affect insect pests in other ways (decreased biting, decreased reproductive output, etc.) that reduce their harmful impact. If that is an argument the authors are putting forth, then it might be more impactful if the argument is stated more directly. It might also be strengthened with modelling or other work demonstrating the quantitative impact of prallethrin on, for example, the vectorial capacity of exposed mosquito populations. If the authors are not putting forth that argument, then some of the text in the Introduction detailing the European Biocidal Product Regulation and EU efficacy requirements can be removed, as some of that detail is not relevant to the immediate study.

SPECIFIC COMMENTS

TITLE/ABSTRACT:

  1. I guess “three-pronged” in the title refers to assays on A) mosquito fitness, B) mosquito biting behaviour, and C) human/environmental health? It’s not clear from the text.

INTRODUCTION:

  1. Lines 64-89. Some of the regulatory detail in these paragraphs is only useful if the authors are making a wider argument in favour of broadening EU guidelines to include active substances with sublethal effects. Otherwise it can be removed.
  2. Prallethrin has apparently been around since the 1980s. Why the interest now? Has it never been considered as a mosquito control agent before?
  3. Is there any known prallethrin resistance in Aedes albopictus or Culex pipiens?

MATERIALS AND METHODS:

  1. Line 114: Most insecticide bioassays expose adults 7 days or younger. Why use adults aged 12-14 days? I would expect higher mortality for older mosquitoes.
  2. To be clear, were the male mosquitoes also 12-14 days old? That would be near the end of their natural lifespan, no?
  3. Lines 127-128: Entirely untreated mosquitoes is not a positive control. A positive control would be a treatment that should produce the observable effect (e.g. death). I’d suggest that the entirely untreated mosquitoes are the negative control, and the mosquitoes exposed to solvent only would be the sham treatment.
  4. Lines 139-142. I’d like to know more about the Culex pipiens strain. Are they Culex pipiens pipiens, or Culex pipiens biotype molestus? The fact that they are autogenous suggests molestus. Is there a reference from a previous paper where the colony was founded or characterised?
  5. Lines 159-161. “Any mortality during this period was noted, and the chamber was considered to be contaminated or in an unsatisfactory state if KD was higher than 10%”. Given that the study was designed to measure sublethal effects, should mosquitoes placed in the chamber pre-trial be checked for sublethal effects as well? A low level of residue in the chamber that caused sublethal effects but not KD would aversely effect results.
  6. Lines 193-198. More detail on the larval rearing process needs to be clarified. How much water was placed in each 5L trap (I assume not 5L, as they would spill easily…)?
  7. What was the temperature range in the rearing room (was it exactly 25C all day, or plus or minus something)?
  8. What brand of rat food?   What quantity of rat food?
  9. It’s interesting that the Culex and the Aedes species were fed the same type and amount of food. In my admittedly anecdotal experience, Culex pipiens larvae were given larger quantities of food than Aedes albopictus. Carrieri et al 2003 Environ Entomol 32(6): 1313-1321 suggests this by demonstrating quantitatively that Culex pipiens is less efficient at transforming food into biomass. Given the importance of diet to physiological outcomes such as response to insecticides, do you think feeding the two species the same type and amount of food would have a differential effect on the two species?
  10. How many larvae per tray?
  11. Lines 218-219. It’s a little concerning that the Culex pipiens were not biting. Given that they are normally nocturnal feeders, was the lighting schedule adjusted so that the mosquito biting was observed during “night”? What was the lighting scheduling for the room?
  12. 298-300 Were pairwise comparisons controlled via Tukey or whichever method was appropriate?

RESULTS

  1. Table 3 and Table 4. Were the pairwise comparisons controlled?
  2. Lines 348-349 “The most obvious symptom was the partial or complete loss of legs…” Surely the most obvious symptom was death?
  3. Lines 447-449. How do you know the females died from drowning? Could they have died from other causes, then fell into the water?

DISCUSSION

  1. Both mosquito species in this study have been in laboratory colony for years, so have almost certainly experienced inbreeding. How applicable are results here to wild populations?

Author Response

We would like to thank the reviewers for their extremely constructive criticism, and we have tried to incorporate their feedback to the greatest degree possible. Within the manuscript, any changes are underlined and highlighted in red.

Authors' Responses please see the attached.

Round 2

Reviewer 1 Report

I am satisfied with the responses to my comments in the original review. The manuscript should be checked once more for some minor errors at the insertion points of the new material:

e.g.  Line 164. A mosquito was considered to be KDIf no contamination was detected (space needed and capital letter to be changed).

Line 203. The trays were checked  every day and additional food is added as needed. (keep tense consistent by changing 'is' to 'was')

Author Response

We again thank the reviewer for taking the time to review our manuscript. Below, in blue, are our responses to the reviewers’ comments.

Line 164. A mosquito was considered to be KDIf no contamination was detected (space needed and capital letter to be changed).

Sorry about this issue. The text now reads "A mosquito was considered to be KD if it was lying on its back and was unable to upright itself. If no contamination was detected..."

Line 203. The trays were checked  every day and additional food is added as needed. (keep tense consistent by changing 'is' to 'was')

We thank the reviewer for pointing this out, the text has been modified.  

Reviewer 2 Report

I am mostly satisfied by the authors' revisions.  A few minor points below.

Lines 129-130.  As discussed, the use of the term "positive control" is not accurate here.  I understand the utility of both types of control described by the authors, but neither fits the descriptor "positive control". I would advise using a different term.

Lines 145-147  I understand why the authors used older mosquitoes; I think it would be good if there was an explanation somewhere that the older mosquitoes would be used because they fed more consistently than younger adults.

Lines 163-164.  There's a spacing error here with some of the corrected text, and possible some characters missing.

Line 159-167.  There's something wrong with the text here.  I think some text was pasted into the wrong place?  Anyway, I will try to rephrase my comment from review #1.  If there was a low level of contamination that did not affect mortality or KD, but did have sublethal effects, then checking for contamination by only checking for mortality/KD would not detect that level of contamination.  Admittedly, it would be difficult to test for sublethal effects, as those tend to be quantitative, not qualitative.  That might be worth mentioning in the Discussion. 

Author Response

We again thank the reviewer for taking the time to review our manuscript. Below, in blue, are our detailed responses to the reviewers’ comments.

Lines 129-130.  As discussed, the use of the term "positive control" is not accurate here.  I understand the utility of both types of control described by the authors, but neither fits the descriptor "positive control". I would advise using a different term.

Thank you. We now use the term “untreated control” instead of “positive control”. The whole paper, including the figures, tables, and graphical abstract, has been modified to incorporate the new terminology.

Lines 145-147 I understand why the authors used older mosquitoes; I think it would be good if there was an explanation somewhere that the older mosquitoes would be used because they fed more consistently than younger adults.

We agree with the reviewer. The text now reads “All the experiments were conducted using 12- to 14-day-old mosquitoes. Although it is standard to estimate mortality in bioassays using mosquitoes of 5–10 days in age, older mosquitoes are more appropriate when changes in biting behaviour need to be evaluated. Thus, mosquito age was standardised for the whole study.”

Lines 163-164.  There's a spacing error here with some of the corrected text, and possible some characters missing.

Thank you for noticing. The typo has been corrected.

Line 159-167.  There's something wrong with the text here.  I think some text was pasted into the wrong place?  Anyway, I will try to rephrase my comment from review #1.  If there was a low level of contamination that did not affect mortality or KD, but did have sublethal effects, then checking for contamination by only checking for mortality/KD would not detect that level of contamination.  Admittedly, it would be difficult to test for sublethal effects, as those tend to be quantitative, not qualitative.  That might be worth mentioning in the Discussion

Thank you for this feedback. The text has been revised and now reads Any mortality or KD during this period was noted, and the chamber was considered to be contaminated or in an unsatisfactory state if KD was higher than 10% [30]. A mosquito was considered to be KD if it was lying on its back and was unable to upright itself [31]. If no contamination was detected, the first set of mosquitoes was removed, and the experimental set of 125 mosquitoes was released to initiate testing. These latter mosquitoes were given 30 min to acclimate to the chamber and were also provided with a piece of cotton wool soaked in a 10% sugar solution (…)”.

We agree that it can be complicated to evaluate sublethal effects. It is for this reason that we performed various controls as part of our study. Every day, the chambers are properly cleaned and, before any experiment is begun,  a pre-control trial is conducted to assess potential low-level contamination. We agree that there could be even more subtle effects that would not be captured in mortality or KD during the pre-control period. To address these potential effects, we employed the untreated control and the negative control, which allowed comparisons with the treatments that extended beyond KD/short-term mortality. Additionally, we assessed biting behavior as part of the study. This variable is extremely sensitive to contamination, even low levels of contamination that are not necessarily reflected in KD. We understand the reviewer’s doubts. However, we also wish to state that ensuring proper chamber decontamination is an essential part of our daily work. We also respectfully feel that our controls properly guaranteed the reliability of our results.

Thank you in advance for your time and consideration,